# Assessment of Accuracy in Unmanned Aerial Vehicle (UAV) Pose Estimation with the REAL-Time Kinematic (RTK) Method on the Example of DJI Matrice 300 RTK

**DOI:** 10.3390/s23042092

**Published:** 2023-02-13

**Authors:** Szymon Czyża, Karol Szuniewicz, Kamil Kowalczyk, Andrzej Dumalski, Michał Ogrodniczak, Łukasz Zieleniewicz

**Affiliations:** 1Department of Geoinformation and Cartography, Faculty of Geoengineering, University of Warmia and Mazury in Olsztyn, Prawocheńskiego 15, 10-720 Olsztyn, Poland; 2Department of Geodesy, Faculty of Geoengineering, University of Warmia and Mazury in Olsztyn, Prawocheńskiego 15, 10-720 Olsztyn, Poland; 3Independent Researcher, 10-051 Olsztyn, Poland

**Keywords:** GNSS, real-time kinematic, unmanned aerial vehicle, total station

## Abstract

The growing possibilities offered by unmanned aerial vehicles (UAV) in many areas of life, in particular in automatic data acquisition, spur the search for new methods to improve the accuracy and effectiveness of the acquired information. This study was undertaken on the assumption that modern navigation receivers equipped with real-time kinematic positioning software and integrated with UAVs can considerably improve the accuracy of photogrammetric measurements. The research hypothesis was verified during field measurements with the use of a popular Enterprise series drone. The problems associated with accurate UAV pose estimation were identified. The main aim of the study was to perform a qualitative assessment of the pose estimation accuracy of a UAV equipped with a GNSS RTK receiver. A test procedure comprising three field experiments was designed to achieve the above research goal: an analysis of the stability of absolute pose estimation when the UAV is hovering over a point, and analyses of UAV pose estimation during flight along a predefined trajectory and during continuous flight without waypoints. The tests were conducted in a designated research area. The results were verified based on direct tachometric measurements. The qualitative assessment was performed with the use of statistical methods. The study demonstrated that in a state of apparent stability, horizontal deviations of around 0.02 m occurred at low altitudes and increased with a rise in altitude. Mission type significantly influences pose estimation accuracy over waypoints. The results were used to verify the accuracy of the UAV’s pose estimation and to identify factors that affect the pose estimation accuracy of an UAV equipped with a GNSS RTK receiver. The present findings provide valuable input for developing a new method to improve the accuracy of measurements performed with the use of UAVs.

## 1. Introduction

Unmanned aerial vehicles (UAVs) play an increasingly important role in the modern world due to the growing demand for data acquisition services and service automation. Advanced applications are permeating every area of human life, and they stimulate the development of new UAV technologies [1]. Due to their widespread availability and ease of use, unmanned aircraft systems drive the development of new military technologies, georeferencing applications, transport and environmental protection solutions [2,3,4,5,6,7,8]. An analysis of current military conflicts around the world indicates that the use of unmanned vehicle systems in air and on land has profound implications for combat. In addition to surveillance and reconnaissance missions, UAVs can be also used to stage attacks on the enemy, gather information about enemy forces, and gain advantage in the battlefield. The military role of UAVs is growing at an unprecedented rate, and military technologies also drive the development of civilian applications.

Comprehensive UAV solutions offered by drone manufacturers, such as DJI, Autel and Yuneec, can be configured to perform measurements at low altitude. Due to their popularity and reduced cost, commercial UAV technologies have become more accessible to various industries. High market competition and the emergence of specialized UAV systems have contributed to the wide use of UAVs in mapping and surveying. Unmanned vehicle systems have a growing number of applications, and new components are being introduced to facilitate UAV operation and promote the development of new technologies. The existing flight control and support solutions, new communication and data transmission technologies, and new data sensors clearly indicate that UAVs play an increasingly important role in industry and in everyday tasks [9,10].

A review of the literature indicates that UAVs are widely applied to generate accurate data and perform complex engineering tasks in geodesy [11], surveying, photogrammetry, and remote sensing [12,13], for mapping wetlands [14], for Cultural Heritage surveys [15,16], in urban change detection [17], to inspect and control objects that are difficult to access due to their location (bridges) and height (tall buildings, chimneys, power transmission towers) [18,19,20], to control photovoltaic panels with the use of thermal imaging cameras [21], to conduct firefighting and police operations, search for missing persons, and perform various tasks in agriculture [22,23,24] and forestry [25,26].

The growing popularity of UAV solutions for commercial, recreational, and private use has led to the development of the U-Space ecosystem based on the Unmanned Traffic Management (UTM) concept [27]. The U-Space concept was implemented by the regulatory authorities of the European Union and the EU Member States [28]. The main purpose of U-Space is to integrate manned and unmanned aviation, promote the development of UAV services, and ensure safe air navigation. The system supports the coordination of UAV flights, air traffic management, and facilitates the procedures for UAV users. The U-Space concept has been introduced to cater to the rapid development of UAV services and to adapt the existing procedures to the advanced capabilities of modern drones. In this respect, U-Space creates new administrative and legal options for implementing automatic and autonomous aircraft [21,29,30]. In regard to automatic flights, various procedures are developed and uploaded to a drone’s onboard computer to facilitate UAV operation. Automatic procedures are implemented mainly to maintain the direction of flight along a predefined trajectory, control flight speed, detect and track an object, and execute the Return to Home (RTH) command when the drone loses contact with the operator. Control processes and reliable flight control software during autonomous flights play a very important role in research on advancements in UAV technology.

Reliable navigation is the key prerequisite in the development of automated UAV systems. Modern drones are equipped with many sensors that enable the operator to accurately determine their position and increase the reliability of navigation. In view of the above, further research is needed to promote the development of reliable methods for UAV pose estimation, and to integrate these technologies with a drone’s external sensors. This is a highly challenging task, as demonstrated by the studies undertaken to verify the accuracy of Global Navigation Satellite System (GNSS) receivers equipped with real-time kinematic (RTK) positioning systems and Inertial Measurement Unit (IMU) devices [31,32,33,34,35]. The pose estimation accuracy of commercial UAV solutions continues to increase. Technological advances, the development of new components, and their compatibility and integration necessitate further research into positioning accuracy, including with the use of traditional methods that rely on terrestrial equipment to determine the coordinates of survey points [36,37].

The undertaken research problem has significant implications for fully automatic UAVs that are flown without the operator’s involvement based on a programmed procedure. The problem of drone flight planning under uncertainty is also an interesting research problem for the Artificial Intelligence and Robotics community, as well as the Dynamic Systems and Controls community [38]. The problems associated with ensuring flight stability and safety were classified by La Valle and Sharma [39,40] as:Uncertainty in vehicle dynamics and limited precision in following commands,Uncertainty in knowledge of the environment, including obstacles in it,Unpredictability of factors in operating environment,Uncertainty of position information.

The importance of the above planning and flight safety issues for UAV increases due to their speed, more complex dynamics and limited payload carrying capability [38].

Therefore, the authors decided to focus on the issue of uncertainty of position information, which is particularly important in terms of the precision of the data acquired. In this aspect, the issue of integration of the sensors carried by the drone, GNSS receivers and IMU and their spatial orientation is relevant [10]. Other different solutions are being analyzed to improve the ability to determine an accurate position, particularly in the unknown environment of drone operation. For reasons of availability, most of the research focuses on solutions based on GPS-RTK, extended with additional supporting systems. The basis for research in this area is the deficiencies of the RTK method, concerning the inability to secure continuous network stability and the failures of the components that are parts of the RTK-GPS system [41,42]. For this reason, in the case of accurate measurements, a study comparing RTK and PPK technologies concluded that the lowest error rate of the processed data was obtained using the PPK-GNSS method [43]. Nevertheless, considering the speed of data acquisition and uncomplicated nature of the RTK method, as well as the accuracy parameters assumed in the model specifications, it is most often used in surveying. Therefore, considering the demands of manufacturers to abandon Ground Control Points (GCP), the problem of the accuracy of the GNSS RTK method became the subject of this study.

The problems presented above with the use of GNSS RTK technology in new drone applications, whose limitations relate to indoor operation as well as instability due to multipath or jamming, lead researchers towards alternative solutions for accurate positioning. In an attempt to solve the problems encountered, research has been conducted into increasing the number of GPS receivers used in relation to a single drone [44], as well as integrating RTK GNSS calculation software with the drone module, which would increase the precision of the positioning through transmitted corrections [45,46]. As part of research on improving the accuracy of position determination by propagating the RTCM messages from the Ground Control Station (GCS), scientists proposed mesh network involving multiple drones [41,47]. In the perspective of further research, it has been proposed to use one of many drones, which would also be located in the area of operation, as a GCS transmitting corrections.

The identification of new solutions to minimize the problem of uncertainty in UAV position information is being pursued not only through improvements to the GNSS RTK method itself, but also by considering other methods of acquiring position information [39,40]. The solution is to acquire data from other sensors, which, in the case of large drones and their extended payloads, can be completed in an extensive set [48,49]. Prototype solutions of fully autonomous UAVs have been proposed, and they can be expected to support decision making in the future. Simultaneous Localization and Mapping (SLAM) algorithms are one of the solutions that can drive the development of fully autonomous drones. SLAM algorithms support precise identification of objects while simultaneously keeping track of the drone’s position within the environment to enable accurate navigation [50,51,52,53,54]. However, this method, has some limitations due to self-repeating patterns as well as use in poor (night) and dynamically changing light conditions [10]. Due to the limitations of the method, the use of the SLAM algorithm is not the only solution that has been investigated for precise drone positioning. Among the basic visual odometry methods used in the field of drone positioning, we can mention solutions using algorithms such as SfM (Structure from Motion) [55,56], PTAM (Parallel Tracking and Mapping) [57] and DTAM (Dense Tracking and Mapping) [58]. Another approach is to develop stand-alone systems such as AHRS (Attitude Heading Reference System) interacting with the Internal Measurement Unit (IMU) [59]. Another alternative to the GNSS-RTK system is Ultra-Wide Band (UWB) distance sensors [60] or pseudolites [61,62].

The main advantage of autonomous UAVs is that decisions concerning flying maneuvers, sensor activation, or the use of additional equipment are not programmed in advance but are made independently by the machine based on real-life circumstances. The present study was undertaken to assess the accuracy of UAV navigation systems, and the results can significantly contribute to the development of autonomous aircraft.

Currently, it should be noted that in commercial applications, especially for the low-budget UAV segment, GNSS-RTK technology is the optimal choice. The authors, considering the above-mentioned problems in used method and the accuracy parameters specified in the specification of the model, decided to verify the precision of the drone’s positioning through the experiments undertaken. Therefore, the main aim of the research was to verify the stability and accuracy of positioning using the GNSS-RTK method.

## 2. Materials and Methods

The present study was undertaken to compare and assess the accuracy of UAV pose estimation based on the data generated by a drone’s onboard navigation system and data acquired during measurements with two independent tachometers. The measurements were conducted in three scenarios. For the needs of the study, flight missions were planned in a selected area; a reference frame of geodetic control points was designed, stabilized, and measured; waypoints were stabilized, and tachometer stations were installed (Figure 1). Tachometric measurements of the drone’s geographic coordinates were performed with the use of a 360° prism reflector connected to a DJI M300 RTK UAV. In the experiments, authors also considered the influence of atmospheric conditions on the measurement results obtained. The flights therefore took place in a wind that blew from a northerly direction with an average speed of 6 km/h. The Kp indicator, which was 1, was also taken into consideration.

The study analyzed surveys made using the GPS RTK method, as a consequence of using a receiver installed on a drone [63,64,65] and tachometric method [66,67,68], as a reference method for UAV position measurements. The determination of the position of the P-point on the Earth’s surface, within the RTK method, is implemented on the basis of a space resection, using a fast initialization based on the on-the-fly (OTF) method [69]. The calculation of the distance from the satellite to the receiver is based on the combined use of code and phase measurements, where the observation equation can be written as follows Equation (1):(1)$$\Phi +v=\frac{1}{\lambda }\rho \left(X_{c}\right)+N$$
where:

Φ—DD carrier phase observable (in cycles),

λ—signal wavelength,

v—residual (measurement noise),

X_c_—receiver coordinate vector,

$\rho \left(X_{c}\right)$—DD geometrical range,

N—integer number of cycles (DD initial ambiguity).

The tachometric method, which is based on a situational-elevation measurement performed using the polar method and trigonometric levelling, is used to determine the situational position and elevation of points. Plane coordinates relative to the surveying ground points were calculated using Equations (2) and (3):(2)$$X_{P}=X_{S}+d_{\mathrm{SP}}{\cos A}_{\mathrm{SP}}$$
(3)$$Y_{P}=Y_{S}+d_{\mathrm{SP}}{\sin A}_{\mathrm{SP}}$$
where:

$X_{P}$, $Y_{P}$, $X_{S}$, $Y_{S}$—plane coordinates of the points P and S,

$d_{\mathrm{SP}}$—horizontal distance between the points P and S,

$A_{\mathrm{SP}}$—azimuth of the line SP.

In turn, the altitude coordinate H was calculated according to Equations (4) or (5):(4)$$H_{P}=H_{S}+\mathrm{dctgZ}$$
or
(5)$$H_{P}=H_{S}+d\prime \cos Z$$
where:

$H_{P}$, $H_{S}$—altitude coordinates of a point P and S,

d, d′—distance reduced to horizontal or inclined distance,

Z—zenith angle (zenith length).

DJI MATRICE 300 RTK is one of the most popular commercial drone platforms. According to the manufacturer’s specifications, the platform has a net weight of 8.37 kg and dimensions of 810 × 670 × 430 mm (L × W × H). The drone is equipped with a GNSS RTK receiver compatible with GPS Navstar, Glonass, BeiDOU, and Galileo systems. Two GNSS RTK antennas are positioned on the arms, and the drone’s position is determined with the use of virtual reference stations. DJI MATRICE 300 RTK has vertical and horizontal hovering accuracy (windless or breezy) of ±0.1 m in D-RTK mode. For the needs of the study, the UAV was expanded to include a platform for mounting the Leica GRZ122 360° prism reflector (Figure 2).

Geodetic coordinates in the reference frame were determined with the static GNSS method during two measurement sessions lasting 45 min each with 4 TOPCON Hiper SR GNSS receivers (Figure 3). Coordinates were mapped in the PL 2000 National Two-Dimensional Coordinate Reference System. The measurements were linked with the ASG-EUPOS network of permanent reference stations, from which the nearest (ID: OPNT) was located at a distance of 3.7 km from the site of the survey. At the same time, 4 points of the basic altimetric network located in the nearest vicinity, whose distances from the place of flight ranged from 0.8 km to 2.49 km, were used in the measurements to establish the newly measured points. Coordinates calculated from the grid alignment were mapped with a horizontal accuracy of 0.002 m and a vertical accuracy of 0.005 m. The location of tachometer stations using the Free Station method is based on highly accurate coordinate measurements. The drone’s position was verified with the use of Leica Viva TS30 and Leica Viva TS15 robotic tachometers (Figure 3).

The tachometers were synchronized to facilitate a comparison of the measured data. Tachometric measurements were conducted in prism tracking mode to increase sampling frequency. The results were compared based on angle measurements and distance measurements in tracking mode according to Figure 3. The location of tachometers and their coordinates are presented in Figure 4.

In the first scenario, the stability of absolute pose estimation was determined while the UAV hovered above a point with known coordinates. Drone coordinates were determined with a robotic tachometer within a time interval of 1 s. The UAV was programmed to reach a target point (point 1001) with known coordinates, and to maintain its position at three altitudes above the point: 1.5 m, 5 m and 10 m. The measurements were performed in rapid succession at each altitude to ensure consistent weather conditions and to minimize the impact of external factors. Therefore, the stability of the drone’s position could be affected mainly by speed and direction of airflow. The study site and specific lower altitudes were selected to minimize the impact of wind on the conducted measurements. At the same time, as part of the experiment conducted, the dependencies and magnitudes of the prism displacement relative to the drone’s position recorded in the analyzed reports were established. The determined values were then taken into account in the subsequent stages of the research.

In the second experiment, the UAV’s position was analyzed during flight along a predefined trajectory. The flight mission was conducted at an altitude of 10 m along a path defined by the coordinates of mission points. The mission was planned to ensure compliance with the requirements for performing low-altitude photogrammetric surveys. The drone was flown above a square-shaped area, and it hovered and changed direction above the main waypoints. Additional waypoints for capturing images were introduced in each direction of flight. Measurements were performed at 14 waypoints. Three sets of coordinates, including waypoints, the drone’s position determined by the onboard GNSS RTK receiver (AIRDATA), and the drone’s position determined based on tachometer readouts, were analyzed.

The third experiment was a modified version of the second experiment, and it was performed to analyze the UAV’s position during uninterrupted flight without hovering over waypoints. The remaining parameters and assumptions were identical to the second experiment. The second and the third research tasks were based on the solutions applied in low-altitude photogrammetric surveys, and they were designed to assess the accuracy of coordinate estimation relative to the derived product (such as an orthophotomap) in several stages: assessment of land cover, calculation of coordinates at the center of the image, and estimation of the accuracy of the derived products. Terrestrial measurements were conducted in prism tracking and automatic geo-registration mode. A minimum of five measurements in each waypoint were performed by both terrestrial stations. Figure 5 shows the scope and the different stages of the experiments carried out.

## 3. Results

The experiments generated sets of independent coordinate measurements. The first dataset contained information about the UAV’s coordinates based on the data logged by a dedicated web application for DJI solutions. The AIRDATA application registers flight parameters, weather conditions, battery levels, and displays alerts about equipment and environmental factors that could impact flight performance (Figure 6). Coordinates were registered in the decimal system (WGS 84), and they had to be converted to the PL 2000 reference format for comparison with tachometer data. The UAV registered its position ten times per 1 s. The second dataset comprised tachometer measurements.

In the first experiment, the drone’s position was determined only with the use of the Leica Viva TS15 instrument. Both tachometers were used to register the drone’s position during the performance of specific tasks. In each experiment, the coordinates of mission points were determined with a dedicated UAV application. Measurement times were synchronized, and the drone’s coordinates, both terrestrial and those determined by RTK GNSS, were visualized in QGIS software.

The results of the first experiment were presented in tabular form of statistics for X- and Y-coordinates in the PL-2000 reference system at the tested altitudes, and as differences between the coordinates measured by a tachometer in the eastern part of the study site (K) and the coordinates measured by the UAV (L) (Table 1, Table 2 and Table 3).

A comparison of the coordinates measured at each examined altitude revealed that the measurements conducted at H = 1.5 m should be additionally verified. For this purpose, 20 additional tachometer measurements (K) were conducted over a period of 36 s. To facilitate data comparison, the coordinates registered by the UAV (10 measurements per second) were averaged for a 1-s interval. The determined values were then compared with the model values used in the UAV mission (M).

The purpose of the second and the third experiment was to generate additional data to verify the accuracy of the UAV’s pose estimation during a planned mission. During these experiments, the drone’s coordinates were measured at three altitudes with a sampling frequency of five measurements per 10 s (Figure 7). Similar to the first experiment, data registered by the drone were acquired via the AIRDATA application.

The standard deviation was below 0.01 m for coordinates measured at flight altitudes of 1.5 and 5 m. This parameter was considerably higher at the altitude of 10 m, and it reached 0.05 m for the X-coordinate (Northing) and 0.02 m for the Y-coordinate (Easting). The differences between tachometric and UAV measurements ranged from 0.04 to 0.07 m for the X-coordinate, and from 0.01 to 0.04 m for the Y-coordinate. The error in the UAV coordinate estimation was 0.06 m for the X-axis and 0.01 m for the Y-axis. The error in UAV pose estimation was 0.06 m at altitudes of 1.5 m and 5 m, and 0.05 m at the altitude of 10 m.

The spatial distribution of UAV pose states is presented in the diagram below (Figure 8). Circles denote buffer zones with a radius of 0.05 m, 0.10 m, and 0.20 m, and present the distance from the waypoints set up in the mission plan. The drone’s position measured with a tachometer is marked in green (K), and the coordinates acquired from the UAV are marked in red (L). The analysis revealed that at an altitude of 10 m, the drone’s coordinates were shifted by 0.10 m to 0.20 m relative to theoretical coordinates. This difference did not exceed 0.10 m at the remaining altitudes (1.5 m and 5 m).

In the second experiment, the differences between model data (coordinates) introduced to the mission (model values, M), the results of terrestrial measurements conducted by two tachometers (K and A), and the coordinates registered by the UAV (L) were compared. The average differences between waypoint values and the measured values are presented in diagrams (Figure 9). It should be noted that at least five terrestrial observations were performed at each waypoint. The data acquired from the UAV via the AIRDATA application were processed in the same manner as in the first experiment.

The scatterplots presenting the standard deviation for every measuring device revealed low data spread. On the X-axis, the standard deviation of the measured values ranged from 0.01 m to 0.04 m for most points. Standard deviation values were two or three times higher only in sporadic cases (Figure 10). The measurements of the Y-coordinate were more stable, and standard deviation ranged from 0.01 m to 0.07 m for all waypoints.

The UAV’s flight trajectory relative to ground control points is presented in Figure 11. The values for the entire mission are marked in black, and the values determined during theoretical hovering above a waypoint are market in red. Buffer zones with radii of 0.05 m, 0.10 m and 0.20 m, describing the distance between the UAV’s position and the planned waypoints, are also presented in the diagram. Differences in the centering accuracy, which is independent of the direction of flight and changes in direction, can be also observed relative to all waypoints.

Most of the analyzed positions were determined within the 0.10 m buffer zone. In regard to points No. 1, 5 and 10, a large number of the identified positions did not fall within the 0.05 m buffer zone. At point No. 10, the distance between the positions determined by the UAV and the waypoint ranged from 0.10 to 0.20 m.

The purpose of the third experiment (scenario) was to conduct a photogrammetry mission during which photographs were taken. This experiment was conducted to verify the precision with which the UAV’s position was recorded (in a continuous manner) during the performance of a standard measurement task. Selected measurements are presented in the diagram to illustrate the variability in X- and Y- coordinates determined with the analyzed measuring techniques (Figure 12). Due to a large dataset and the presence of outliers, the diagram presents measurements conducted over a period of approximately 200 s.

Based on the results of the second experiment, which compared the modeled values with tachometric measurements and GNSS RTK measurements during the flight, in the next stage of the study, an attempt was made to compare the differences between UAV coordinates and terrestrial measurements. During photogrammetric measurements conducted for surveying purposes, images are captured by drone cameras, and the obtained results and their evaluation have the most important practical implications. The results of the comparison, including the values of X- and Y-coordinates, are presented in the below diagram (Figure 13). The diagram shows variations in the differences between the coordinates measured by the UAV and registered in the AIRDATA application (L) and the coordinates measured by tachometers (A) and (K). Data are registered continuously in the AIRDATA application, and the time marking the beginning of the mission was set at 7800. The third experiment lasted 120 s due to the size of the study area. The difference in the values of coordinates determined by both tachometers relative to the UAV’s position approximated 0.05 m for both X- and Y-coordinates.

Similar to the first and second experiment, the results generated by the third experiment were compared with the projected flight path (green solid line). The actual flight trajectory is presented relative to the planned waypoints in the below diagrams. To improve legibility, the measured values were presented on a larger scale for selected segments of the flight path near waypoints 3, 6 and 9 relative to the flight plan presented in the below Figure 14. Similar to the first and second experiment, the diameter of the buffer zones was set at 0.05 m, 0.10 m and 0.20 m to better illustrate the scatter of coordinates.

The analyses revealed that the difference between most positions determined by the UAV and tachometric measurements relative to the planned flight path did not exceed 0.05 m. Greater differences were observed in sporadic cases, but they did not exceed 0.10 m. The most important finding of the study is that the compared measurements did not differ by more than 0.10 m.

## 4. Discussion

The first experiment was based on a representative sample of measurements conducted in a hovering position. The accuracy with which coordinates were registered by the UAV was assessed based on an analysis of descriptive statistics. The difference between the values of X- and Y-coordinates in a 2D system, determined with the use of selected measurement methods, reached 0.02 m. Standard deviation exceeded 0.05 m only minimally, which indicates that the GNSS RTK technology supports highly accurate UAV pose estimation. In the remaining two tasks of experiment 1, which involved changes in the UAV’s hovering height, greater differences in the measured values and standard deviation were noted for both coordinates. At altitudes of 5 m and 10 m, the above statistics increased and exceeded the boundary value of 0.10 m, and standard deviation was determined at 0.05 m for the X-coordinate and 0.02 m for the Y-coordinate. However, fewer observations were made during both tasks than in the first stage of the experiment; therefore, the sample could have been too small. Despite the above, these results suggest that an increase in height can compromise the accuracy of measurements. According to the authors, measurement accuracy was undermined mainly by external factors, in particular weather conditions. In field surveys, the impact of these factors can be minimized, but never completely eliminated. However, with significantly higher heights above 10 m, the impact of atmospheric factors is expected to be greater. These observations were validated by simultaneous terrestrial measurements which revealed similar changes in the monitored values, as presented in the below table (Table 4).

The second and the third experiment confirmed the hypothesis that a significant correlation exists between a UAV’s position determined with the GNSS RTK receiver, tachometric measurements, and the adopted coordinates of waypoints in each mission. These experiments also proved that centering accuracy in selected points is not influenced by the direction of flight or changes in direction. The results of experiments 2 and 3 revealed that the measurements conducted during uninterrupted flight over waypoints were more consistent with tachometric measurements. In the second experiment, where the drone was programmed to hover over each waypoint, flight coordinates were determined by the UAV less accurately relative to the programmed flight path.

The analysis of the obtained results presented in Table 5 allows to conclude that if the hovering is longer over a point with specified coordinates, the difference is smaller between the total station measurement and the planned waypoints. In the second experiment, which involved stopping the drone over a point, we can see many times larger differences in the range between the tachometric measurement and the model data. Of course, this is also influenced by the fact that in experiment no. 2 we analyzed the data for 14 points, while in first experiment measurements only concern one point.

Analyzing the problems encountered with the accuracy of the GNSS-RTK receiver placed on the drone, it should be noted that the measurements were made in open terrain. Therefore, the acquisition of spatial data for areas with a much more complex spatial structure, such as built-up, industrial and mountainous areas, will be associated with lower stability of the used correction method. As a result of inaccurate positioning of the drone during spatial data acquisition in a complex environment, there is a risk of incomplete data acquisition for the entire area, as well as incorrect realization of the assumed lateral and longitudinal coverage.

Another conclusion of the study is the need to use GCPs to assess the quality and verify the data acquired from the aerial survey. The GCP, especially in the case of complex areas, should be regularly spaced, which is supported by the values shown in Table 5. It is also important to point out that GPC points also solve the problem of establishing the correct heights of the area for which the data is acquired. The authors believe that accurate elevation data acquisition using UAVs is a challenge that they would like to analyze in further research.

Authors believe that research into the accuracy of using the GNSS-RTK method for UAVs should be continued due to its speed and ease of method application. However, it should be noted that this method is not without its drawbacks, which are related to the possibility of losing the connection which sends corrections, particularly in the case of BVLOS (Beyond Visual Line of Sight) flight. It is also problematic, as analyzed in this article, to be able to obtain measurements with centimeter accuracy, especially in the subject of continuous flights without stopping at the time of data collection.

Authors believe that the proposed method, using tachometric measurements as a complement to the UAV’s data collection process, will allow for the accurate determination of the drone’s position, as well as the verification of the accuracy of the measurements taken by the drone’s sensors.

The study also revealed that Y-coordinates should be determined more accurately relative to X-coordinates, and this observation has important implications for further research. In the future, the configuration of flight missions should be modified to minimize the effect of the azimuth on pose estimation accuracy. This aspect should be analyzed in detail to determine whether X- and Y-coordinates are affected by the direction of flight when the mission is adequately configured.

## 5. Conclusions

The accuracy of a UAV’s pose estimation is the key determinant of drones’ applicability for surveying operations. This is an important consideration since despite their limitations, UAVs considerably speed up field surveys. Professional solutions are expensive, but most commercial UAV models are affordable for contractors performing surveying operations.

The study demonstrated that the positioning accuracy of the DJI MATRICE 300 drone equipped with a GNSS RTK receiver is consistent with the manufacturer’s specifications. The GNSS RTK receiver correctly mapped X-coordinates, and the difference between the values measured by the UAV and model data increased with a rise in flight altitude. The conducted experiments revealed that most deviations can be attributed mainly to external factors, including weather conditions. The influence of these factors can be minimized, but not completely eliminated in field surveys. A UAV’s pose estimation accuracy cannot be reliably determined in a laboratory, and field tests should be carried out to verify the system’s capabilities under real-life conditions. Both the direction and altitude of flight significantly affect pose estimation accuracy.

The proposed test procedure was positively validated in the study. Considering the safety of navigation, especially during missions, and the efficiency of data acquisition, this research can provide a basis for developing criteria for evaluating the quality of satellite positioning. A catalogue of features developed on the basis of empirical research would allow the prediction of positioning accuracy problems occurring along the drone’s entire flight trajectory, and in particular at take-off and landing moments.

In the future, the results can be used to determine the effect of waypoint location relative to the direction of the axes in a selected coordinate system during flight missions. In an upcoming study, the authors will analyze the accuracy with which Y-coordinates are determined and the conversion of ellipsoidal height to orthometric height in selected devices.

## Figures and Tables

**Figure 1 sensors-23-02092-f001:**
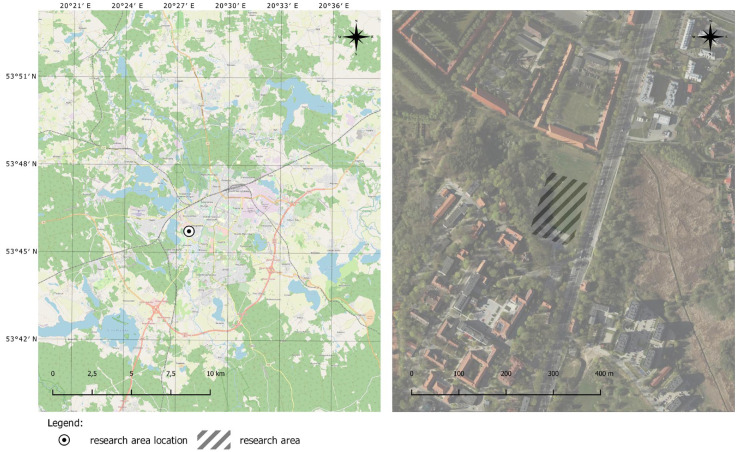
Research area.

**Figure 2 sensors-23-02092-f002:**
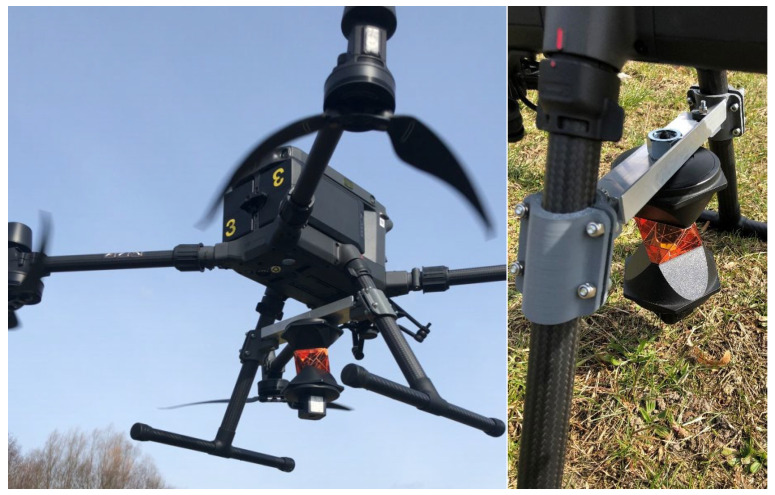
DJI MATRICE 300 RTK with Leica GRZ122 360° prism reflector.

**Figure 3 sensors-23-02092-f003:**
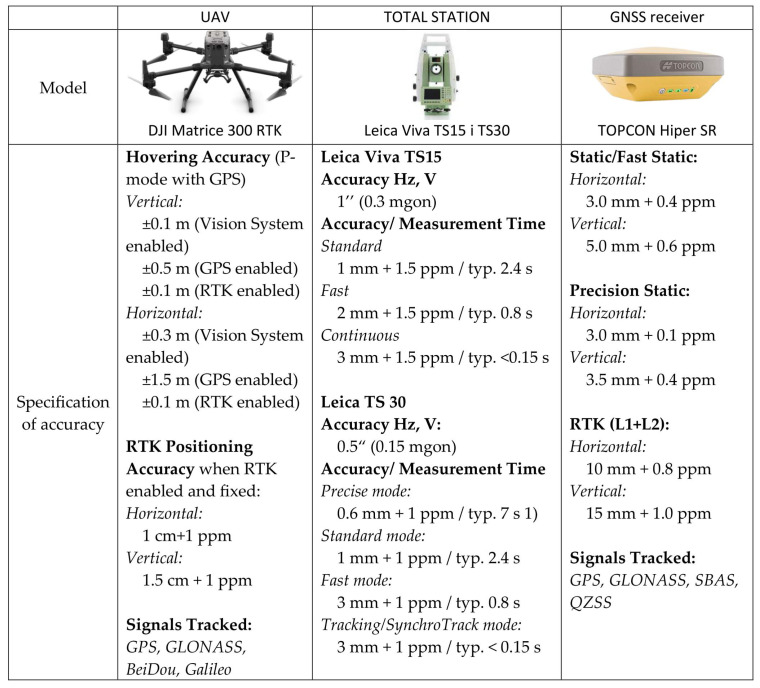
Assessment of the accuracy of used equipment.

**Figure 4 sensors-23-02092-f004:**
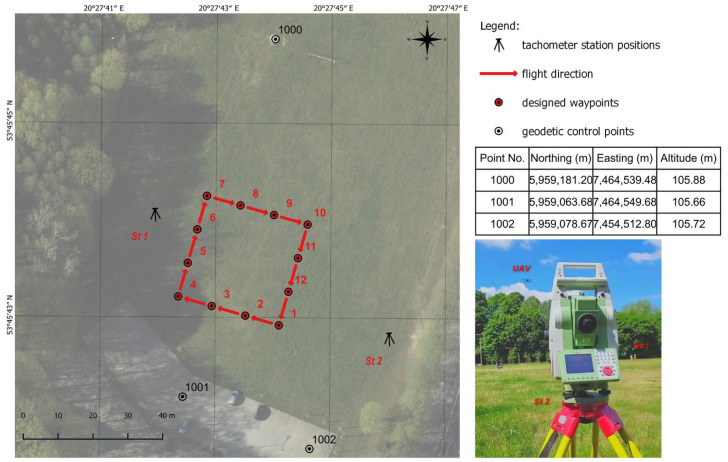
Planned UAV mission.

**Figure 5 sensors-23-02092-f005:**
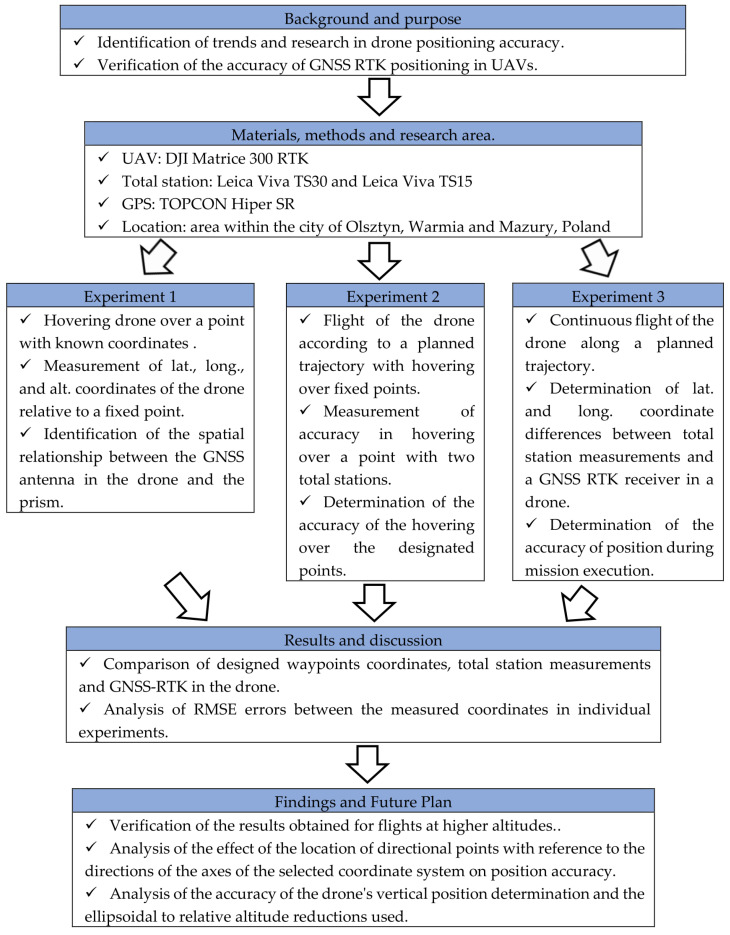
Overview of study procedure.

**Figure 6 sensors-23-02092-f006:**
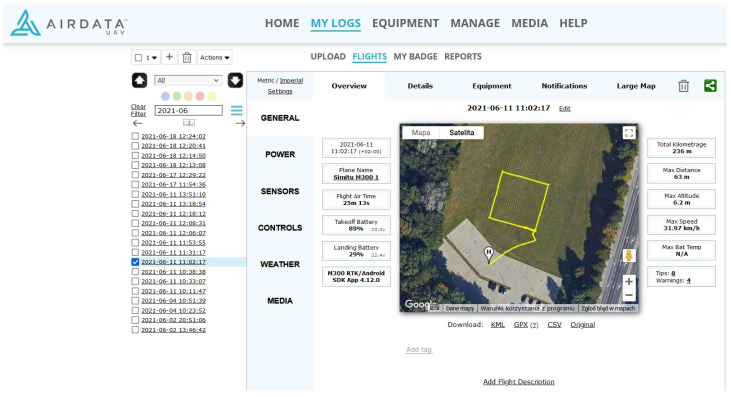
AIRDATA application.

**Figure 7 sensors-23-02092-f007:**
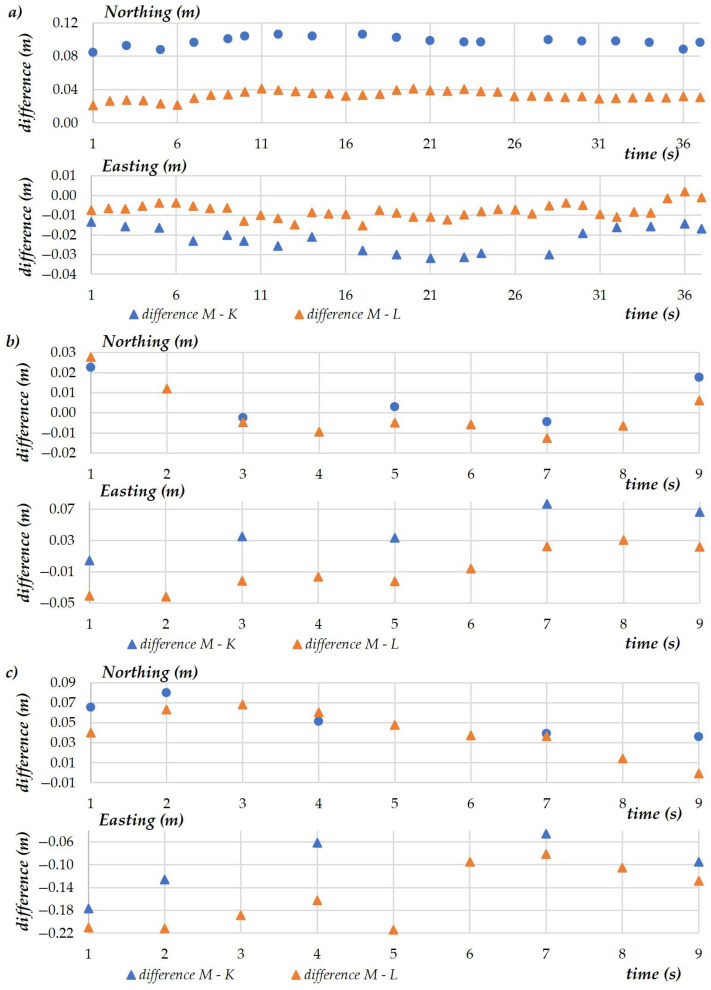
A comparison of the coordinates measured at each examined altitude (**a**) altitude H = 1.5 m; (**b**) altitude H = 5 m; (**c**) altitude H = 10 m.

**Figure 8 sensors-23-02092-f008:**
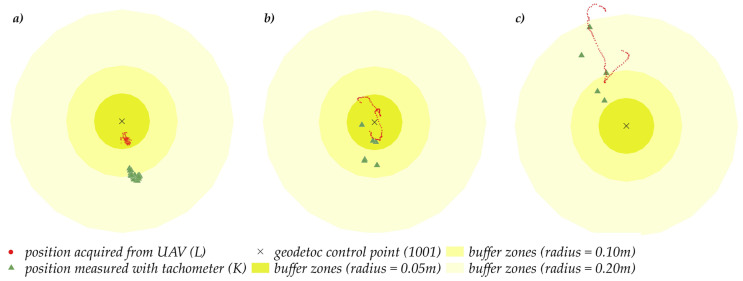
The spatial distribution of UAV positions at each examined altitude (**a**) altitude H = 1.5 m; (**b**) altitude H = 5 m; (**c**) altitude H = 10 m.

**Figure 9 sensors-23-02092-f009:**
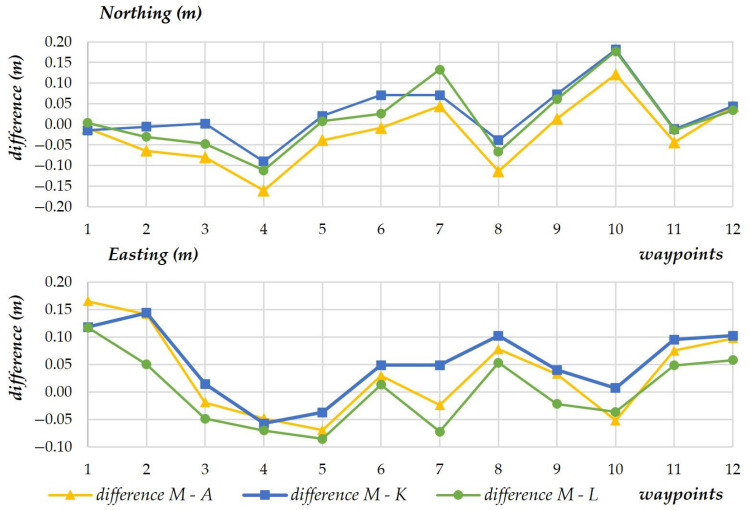
The average differences between model values and the measured values.

**Figure 10 sensors-23-02092-f010:**
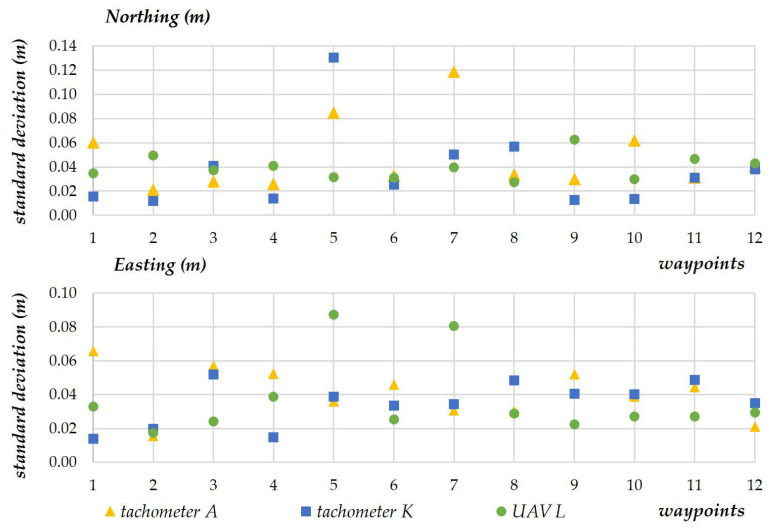
Standard deviation for every measuring device.

**Figure 11 sensors-23-02092-f011:**
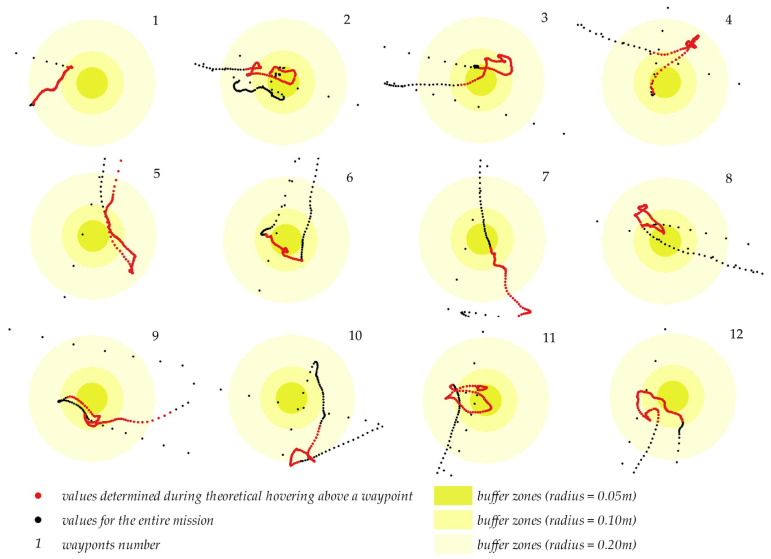
Standard deviation for every observed waypoint.

**Figure 12 sensors-23-02092-f012:**
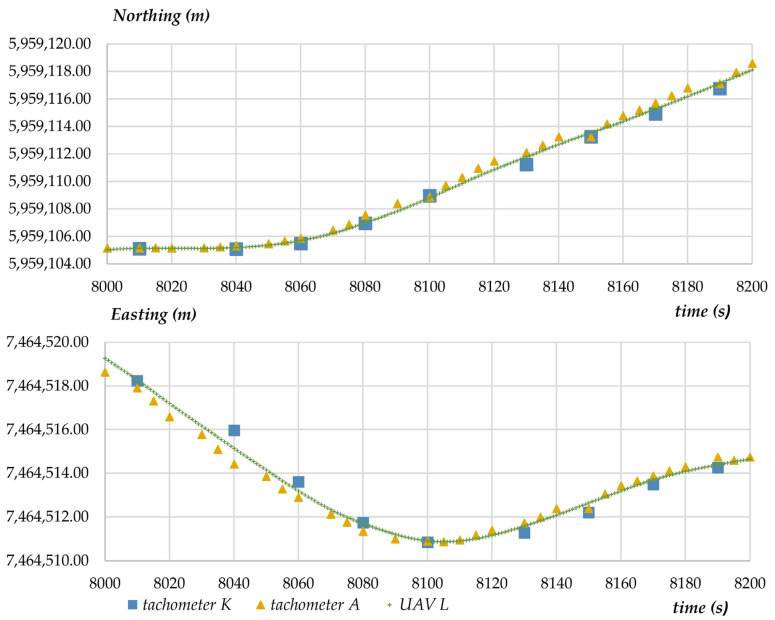
Variability in X- and Y-coordinates determined with the analyzed measuring techniques.

**Figure 13 sensors-23-02092-f013:**
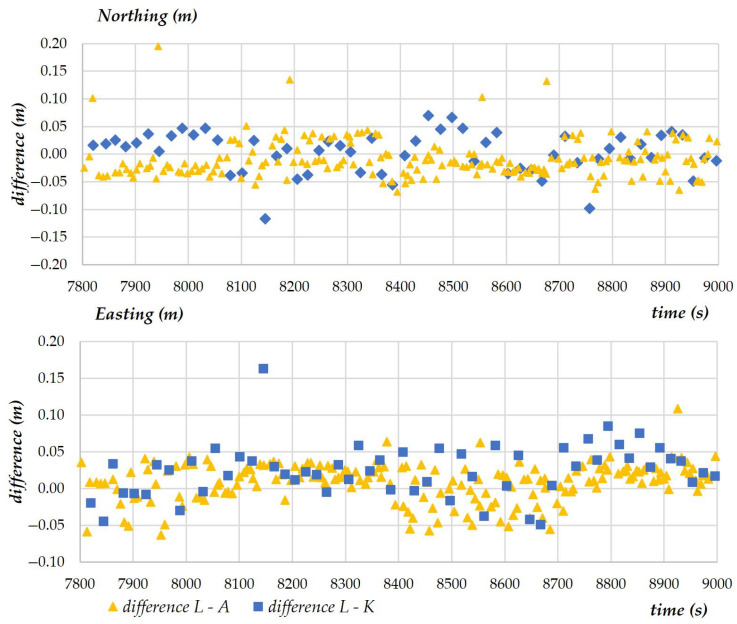
Variations in the differences between the coordinates measured by the UAV and registered in the AIRDATA application (L) and the coordinates measured by tachometers (A) and (K).

**Figure 14 sensors-23-02092-f014:**
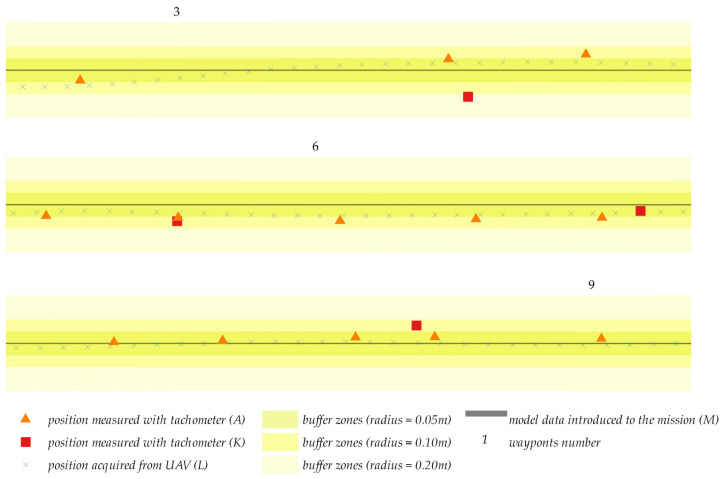
Measured values presented for selected segment of the flight path.

**Table 1 sensors-23-02092-t001:** Statistics for measured coordinates (altitude H = 1.5 m).

	Tachometer K		UAV L		Odds K-L (m)	
	Northing (m)	Easting (m)	Northing (m)	Easting (m)		
Minimum	5,959,078.56	7,464,512.84	5,959,078.63	7,464,512.83	−0.06	0.02
Maximum	5,959,078.59	7,464,512.86	5,959,078.65	7,464,512.85	−0.07	0.02
Mean	5,959,078.57	7,464,512.85	5,959,078.64	7,464,512.84	−0.06	0.01
Std deviation	0.01	0.01	0.01	0.00	−	−

**Table 2 sensors-23-02092-t002:** Statistics for measured coordinates (altitude H = 5 m).

	Tachometer K		UAV L		Odds K-L (m)	
	Northing (m)	Easting (m)	Northing (m)	Easting (m)		
Minimum	5959,078.59	7,464,512.81	5,959,078.64	7,464,512.80	−0.05	0.01
Maximum	5,959,078.64	7,464,512.83	5,959,078.71	7,464,512.84	−0.07	−0.01
Mean	5,959,078.61	7,464,512.82	5,959,078.68	7,464,512.83	−0.06	−0.01
Std deviation	0.05	0.01	0.03	0.01	−	−

**Table 3 sensors-23-02092-t003:** Statistics for measured coordinates (altitude H = 10 m).

	Tachometer K		UAV L		Odds K-L (m)	
	Northing (m)	Easting (m)	Northing (m)	Easting (m)		
Minimum	5,959,078.72	7,464,512.75	5,959,078.75	7,464,512.76	−0.04	−0.01
Maximum	5,959,078.85	7,464,512.79	5,959,078.88	7,464,512.83	−0.04	−0.04
Mean	5,959,078.77	7,464,512.78	5,959,078.83	7,464,512.79	−0.05	−0.01
Std deviation	0.05	0.02	0.05	0.02	−	−

**Table 4 sensors-23-02092-t004:** Statistics for measured coordinates (altitude H = 1.5 m; altitude H = 5 m; altitude H = 10 m).

	UAV (L)		Tachometer (K)	
	Northing (m)	Easting (m)	Northing (m)	Easting (m)
Altitude	1.5 m			
Difference (m)	0.02	0.02	0.02	0.02
Standard deviation (m)	0.01	0.01	0.01	0.00
Altitude	5 m			
Difference (m)	0.04	0.03	0.07	0.04
Standard deviation (m)	0.05	0.01	0.03	0.01
Altitude	10 m			
Difference (m)	0.13	0.04	0.13	0.07
Standard deviation (m)	0.05	0.02	0.05	0.02

**Table 5 sensors-23-02092-t005:** Statistics for measured coordinates in all experiments.

	Experiment 1		Experiment 2		Experiment 3	
	Northing (m)	Easting (m)	Northing (m)	Easting (m)	Northing (m)	Easting (m)
Min difference (m)	−0.050	0.010	−0.090	−0.057	0.010	0.010
Max Difference (m)	−0.070	0.010	0.181	0.165	1.130	1.347
Standard deviation (m)	0.05	0.03	0.040	0.070	0.313	0.357

## Data Availability

Data sharing not applicable.

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
