# Peer review of "Assessment of Accuracy in Unmanned Aerial Vehicle (UAV) Pose Estimation with the REAL-Time Kinematic (RTK) Method on the Example of DJI Matrice 300 RTK"

_sensors, 2023, doi:10.3390/s23042092_

Round 1
Reviewer 1 Report
In this study, a number of experiments are presented to hypothesize that modern navigation receivers equipped with real-time kinematic positioning software and integrated with UAVs can significantly improve the accuracy of photogrammetric measurements. The study touched upon an important research area, especially considering the increasing use of drones with RTK. However, the article needs major revision. Here are my suggestions:
1) The novelty of the study should be emphasized compared to studies in the literature.
2) Explanation about the study should be provided in the last paragraph of the Introduction.
3) Please change the title of Figure 1 to just ‘’Research area’’. Terrestrial measurements are not included in Figure 1.
4) The first two paragraphs of the Discussion section should either be moved to the Conclusion section or be removed altogether. I do not think it is necessary to give general information again in the discussion.
5) Discussion section should be developed. Further discussion should be provided with reference to the tables presented in Section 3.
6) I found the method part quite simple. Mathematics background should be emphasized more. The mathematics of the measurements performed with Total Station and the RTK method on which the article is based should be presented.
7) I could not see any discussion or result for the purpose of increasing the accuracy of photogrammetric measurements claimed at the beginning of the study. Could you explain this part a little more?
Author Response
Response to Reviewer 1 Comments
First of all, we would like to thank you very much for the comments. We are convinced that your suggestions have significantly contributed to an improvement in our article. We do hope that we have succeeded in considering them in a satisfactory manner. Below are answers to specific comments.
1) The novelty of the study should be emphasized compared to studies in the literature.
Suggestion has been inserted into the article.
2) Explanation about the study should be provided in the last paragraph of the Introduction.
Suggestion has been inserted into the article.
3) Please change the title of Figure 1 to just ‘’Research area’’. Terrestrial measurements are not included in Figure 1.
Suggestion has been inserted into the article.
4) The first two paragraphs of the Discussion section should either be moved to the Conclusion section or be removed altogether. I do not think it is necessary to give general information again in the discussion.
Suggestion has been inserted into the article.
5) Discussion section should be developed. Further discussion should be provided with reference to the tables presented in Section 3.
Suggestion has been inserted into the article.
6) I found the method part quite simple. Mathematics background should be emphasized more. The mathematics of the measurements performed with Total Station and the RTK method on which the article is based should be presented.
Suggestion has been inserted into the article.
7) I could not see any discussion or result for the purpose of increasing the accuracy of photogrammetric measurements claimed at the beginning of the study. Could you explain this part a little more?
Suggestion has been inserted into the article in result section.
Submission Date
07 January 2023
Date of this review
11 Jan 2023 20:39:58

Reviewer 2 Report
A brief summary: In the publication, the authors consider and investigate the accuracy of determining the positions of UAVs during flight. Due to the rapid development of UAVs, this area of research is quite relevant and important. The proposed accuracy assessment algorithm is quite new and interesting.
General concept comments:
The introduction presents a general overview of the use of UAVs, but there is no critical analysis of other researchers' assessment of the accuracy of UAV positions. The methodology needs clarification (see Specific comments). The presented results have an extremely short sample of data, for all scenarios. The discussion should be supplemented by a comparison with other studies and critical analysis.
Specific comments
1. It is recommended to make a block diagram of the research methodology, where it is necessary to present the main stages of the research and give a brief description of the scenarios (and possibly measurement schemes).
2. Since the research uses several types of measurements and equipment (GNSS receivers, total stations, prisms360), it is necessary to perform an a posteriori assessment of the accuracy of determining the coordinates of this equipment complex. Because it is obvious that a significant part of the differences you identified in the Results is related to the accuracy of coordinate measurements.
3. It is obvious that meteorological parameters play an important role when determining UAV coordinates. Therefore, it is necessary to specify the values of meteorological parameters during the research (especially wind speed and its direction). And also to analyze their possible influence on the final results.
4. The study used a static method to determine the coordinates of the control points but did not specify such important characteristics as the duration of the static measurements and the distance to the nearest permanent GNSS station(s) (in the ASG-EUPOS network).
5. According to the reviewer, the (planar/horizontal) accuracy of determining the coordinates of control points of 0.002 m specified in the study is overestimated and raises doubts because according to the technical characteristics, the accuracy of the used receiver is 3.0mm+0.1ppm (TOPCON Hiper SR).
6. It is necessary to specify the method by which the coordinates of the total station were determined (probably "FreeStation", but it must be noted).
7. The main condition for conducting the presented research is to ensure the coexistence of the GNSS antenna of the UAV and the attached 360prism. Therefore, it is necessary to provide information about how such coexistence was secured and verified.
8. The implementation of the third scenario is quite debatable, since the total station during tracking may not correctly determine the coordinates of the moving target.
9. To improve the discussion, it is recommended to present a table with the main values of the three scenarios (difference, standard deviation).
Author Response
Response to Reviewer 2 Comments
First of all, we would like to thank you very much for the comments. We are convinced that your suggestions have significantly contributed to an improvement in our article. We do hope that we have succeeded in considering them in a satisfactory manner. Below are answers to specific comments.
Specific comments
- It is recommended to make a block diagram of the research methodology, where it is necessary to present the main stages of the research and give a brief description of the scenarios (and possibly measurement schemes).
Suggestion has been inserted into the article as Figure 5.
- Since the research uses several types of measurements and equipment (GNSS receivers, total stations, prisms360), it is necessary to perform an a posteriori assessment of the accuracy of determining the coordinates of this equipment complex. Because it is obvious that a significant part of the differences you identified in the Results is related to the accuracy of coordinate measurements.
Suggestion has been inserted into the article. Assessment of the accuracy of used equipment shown as Figure 3.
- It is obvious that meteorological parameters play an important role when determining UAV coordinates. Therefore, it is necessary to specify the values of meteorological parameters during the research (especially wind speed and its direction). And also to analyze their possible influence on the final results.
Authors considered the possibility of weather conditions influencing the results. Therefore, the flight took place on a day when the average wind speed was 6 km/h with a northerly wind direction. At the same time, it should be noted that for the first stage of the research, in terms of UAV positioning accuracy, the authors chose lower flight heights. This was due to the fact that the drone would be less affected by atomospheric conditions than flying at higher altitudes. In addition, the Kp index on the day of the experiments was 1.
Above explanations are included in the article in the line 178.
- The study used a static method to determine the coordinates of the control points but did not specify such important characteristics as the duration of the static measurements and the distance to the nearest permanent GNSS station(s) (in the ASG-EUPOS network).
Static measurements lasting 45 minutes were made in order to precisely determine the coordinates of the reference grid points. The nearest permanent GNSS station in ASG-EUPOS network (ID: OPNT) was located at a distance of 3.7 km from the survey site. In the static measurements, 4 points of the basic altimetric network located in the nearest vicinity were used to establish the newly measured points, with the distances from the flight site to the nearest point being 0.8 km and to the furthest point being 2.49 km.
Above explanations are included in the article in the lines 243.
- According to the reviewer, the (planar/horizontal) accuracy of determining the coordinates of control points of 0.002 m specified in the study is overestimated and raises doubts because according to the technical characteristics, the accuracy of the used receiver is 3.0mm+0.1ppm (TOPCON Hiper SR).
Indeed, the accuracy in static measurement of the TOPCON Hiper SR receiver is 3.0mm+0.1ppm. However, this assumes use of dual frequency GPS, precise ephemerides, calm ionospheric conditions, approved antenna calibration, unobstructed visibility above 10 degrees and an observation duration of at least 3 hours (dependent on baseline length). However, for the study in question, the authors used multi GNSS (GPS+GLONASS) and also reduced the horizon cut-off to 5 degrees. In addition, due to the use of altimetric baseline reference points, the error indicated in the article should be considered in the context of grid alignment and not direct independent static measurement.
Above explanations are included in the article in the line 248.
- It is necessary to specify the method by which the coordinates of the total station were determined (probably "FreeStation", but it must be noted).
Suggestion has been inserted into the article.
- The main condition for conducting the presented research is to ensure the coexistence of the GNSS antenna of the UAV and the attached 360prism. Therefore, it is necessary to provide information about how such coexistence was secured and verified.
The main factor determining the location of the prism mount was the possibility of obtaining a stable position during flight, and thus during measurements. The solution adopted allowed the elimination of vibrations that are caused by rotating propellers during flight. Determining the value of the prism's displacement relative to the drone's reported position was part of the first experiment, which were then taken into account in further analyses.
Above explanations are included in the article in the line 286
- The implementation of the third scenario is quite debatable, since the total station during tracking may not correctly determine the coordinates of the moving target.
Authors also identified the risk of the total station losing track of the drone. Therefore, the drone moved at a minimum speed of 6 m/s. In addition, two total station operators monitored at the instruments for correct measurements.
- To improve the discussion, it is recommended to present a table with the main values of the three scenarios (difference, standard deviation).
Suggestion has been inserted into the article.
Submission Date
07 January 2023
Date of this review
08 Jan 2023 22:19:11

Reviewer 3 Report
The manuscript titled “Assessment of accuracy in the unmanned aerial vehicle (UAV) pose estimation with the Real-Time Kinematic (RTK) method on the example of DJI Matrice 300 RTK”. This Manuscript is within the scope of the journal. This kind of study is useful but it lacks proper justification. The authors need to be improved the manuscript and my comments are given below. The authors have employed pose estimation technique with real-time kinematic using the drone DJI Matrices 300 RTK. The authors have employed pose estimation technique with real-time kinematic using the drone DJI Matrices 300 RTK. Authors experimented 1.5-meter, 5 meters and 10 meters, following are my comments on this manuscript:-
Authors experimented 1.5-meter, 5 meters, and 10 meters, Why these 3 heights are selected for the experiment? Generally, the drone flies above more than 50 meters, Than How these results will help in surveying and mapping?
Use of reflector with the landing gear of drone, How it is giving an estimation of the pose of the drone, I need justification behind this.
Why do you want to do this experiment, How this will be useful is not clear in the paper, Give a proper justification.
The experiment must be performed with more than 50-meter height of the UAV.
The methodology is not clear It requires improvement. Also author can give a methodology flowchart so that readers can easily understand the methodology.
Image qualities are very bad, and need to be improved.
The pose of the drone changes when it flies in an auto mission, Why is this experiment for pose estimation?
The results and discussion part is very weak it needs to be improved more.
The Literature review part must be improved to add various applications of UAV as well as other papers strengthening the background of the study. You can take the help of these papers:-.
https://link.springer.com/article/10.1007/s10846-011-9642-9
https://ieeexplore.ieee.org/abstract/document/7487682?casa_token=pZ8290Oi3gEAAAAA:DNceBcCrsafkNhvdDe8PKW1NV6m34Jztm8got-8RjlKCuabhl5P-d1LuFWudTM4zON2O-O_SO0BrI7g
https://www.int-arch-photogramm-remote-sens-spatial-inf-sci.net/XLIII-B1-2022/407/2022/
https://ieeexplore.ieee.org/document/9462495
On the accuracy of position estimation from aerial imagery captured by low-flying UAVs
https://link.springer.com/article/10.1007/s10846-010-9505-9
In a few figures (Fig 5, 7, 8, 10, 11), the axis label color can be darkened to make them more legible.
Author Response
Response to Reviewer 3 Comments
First of all, we would like to thank you very much for the comments. We are convinced that your suggestions have significantly contributed to an improvement in our article. We do hope that we have succeeded in considering them in a satisfactory manner. Below are answers to specific comments.
Authors experimented 1.5-meter, 5 meters, and 10 meters, Why these 3 heights are selected for the experiment? Generally, the drone flies above more than 50 meters, Than How these results will help in surveying and mapping?
Authors decided to carry out the experiment at heights of 1.5 m., 5 m., and 10 m. in order to minimise external factors affecting the accuracy of position measurement. In particular, the key elements that affect the drone at higher altitudes are wind speed and direction. Thank you for your valuable insight and comment. The study is part of a larger research project that will address the issues identified in the review.
Above explanations are included in the article in the line 178.
Use of reflector with the landing gear of drone, How it is giving an estimation of the pose of the drone, I need justification behind this.
The main factor determining the location of the prism mount was the possibility of obtaining a stable position during flight, and thus during measurements. The solution adopted allowed for the elimination of vibrations that are caused by rotating propellers during flight. Determining the displacement of the prism relative to the position of the drone established in the reports was part of the first experiment, which were then taken into account in further analyses.
Why do you want to do this experiment, How this will be useful is not clear in the paper, Give a proper justification.
The analysis of determination accuracy is a key element in surveying work, particularly in the field of surveying. The main objective of the experiment was to verify the accuracy of the use of the RTK method, since, according to the specification, the accuracy determined by the above method for the analysed model of horizontal coordinates is 1 cm + 1 ppm, and of vertical is 1.5 cm + 1 ppm (DJI M300 RTK). In addition, the aim of the authors of the experiment was to develop a procedure for testing the accuracy of commercial UAV models, which are increasingly being used in precision field measurements. The authors assume that they will continue their work related to the development of methods to quickly determine the exact position of a drone.
The experiment must be performed with more than 50-meter height of the UAV.
The research is part of a larger research project that will address the issues identified in the review, including measurement at higher altitudes.
The methodology is not clear It requires improvement. Also author can give a methodology flowchart so that readers can easily understand the methodology.
Suggestion has been inserted into the article.
Image qualities are very bad, and need to be improved.
Suggestion has been inserted into the article.
The pose of the drone changes when it flies in an auto mission, Why is this experiment for pose estimation?
The experiment in question was conducted to verify the position determination of a drone that is in constant motion. Based on experience and previous experiments on the accuracy of GPS RTK receivers used for surveying, it was assumed that this was a less accurate measurement. The aim of the experiment was to determine how large the errors are when the drone moves along a defined trajectory.
The results and discussion part is very weak it needs to be improved more.
Suggestion has been inserted into the article.
The Literature review part must be improved to add various applications of UAV as well as other papers strengthening the background of the study. You can take the help of these papers:
https://link.springer.com/article/10.1007/s10846-011-9642-9
https://ieeexplore.ieee.org/abstract/document/7487682?casa_token=pZ8290Oi3gEAAAAA:DNceBcCrsafkNhvdDe8PKW1NV6m34Jztm8got-8RjlKCuabhl5P-d1LuFWudTM4zON2O-O_SO0BrI7g
https://www.int-arch-photogramm-remote-sens-spatial-inf-sci.net/XLIII-B1-2022/407/2022/
https://ieeexplore.ieee.org/document/9462495
On the accuracy of position estimation from aerial imagery captured by low-flying UAVs
https://link.springer.com/article/10.1007/s10846-010-9505-9
Suggestion has been inserted into the article.
In a few figures (Fig 5, 7, 8, 10, 11), the axis label color can be darkened to make them more legible.
Suggestion has been inserted into the article.
Submission Date
07 January 2023
Date of this review
14 Jan 2023 15:47:18

Round 2
Reviewer 1 Report
Necessary revisions have been made. The Manuscript looks ready for publication.